# Effects of Trehalose Preconditioning on H9C2 Cell Viability and Autophagy Activation in a Model of Donation after Circulatory Death for Heart Transplantation

**Jingwen Gao, Yasushige Shingu \***  **and Satoru Wakasa**

Department of Cardiovascular Surgery, Faculty of Medicine, Graduate School of Medicine, Hokkaido University, Sapporo 060-8638, Japan; jingwen.gao.l2@elms.hokudai.ac.jp (J.G.); wakasa@med.hokudai.ac.jp (S.W.)
\* Correspondence: shingu@huhp.hokudai.ac.jp; Tel.: +81-11-716-1161

**Abstract:** Donation after circulatory death (DCD) is a promising strategy for alleviating donor shortage in heart transplantation. Trehalose, an autophagy inducer, has been shown to be cardioprotective in an ischemia-reperfusion (IR) model; however, its role in IR injury in DCD remains unknown. In the present study, we evaluated the effects of trehalose on cardiomyocyte viability and autophagy activation in a DCD model. In the DCD model, cardiomyocytes (H9C2) were exposed to 1 h warm ischemia, 1 h cold ischemia, and 1 h reperfusion. Trehalose was administered before cold ischemia (preconditioning), during cold ischemia, or during reperfusion. Cell viability was measured using the Cell Counting Kit-8 after treatment with trehalose. Autophagy activation was evaluated by measuring autophagy flux using an autophagy inhibitor, chloroquine, and microtubule-associated protein 1A/1B light chain 3 B (LC3)-II by western blotting. Trehalose administered before the ischemic period (trehalose preconditioning) increased cell viability. The protective effects of trehalose preconditioning on cell viability were negated by chloroquine treatment. Furthermore, trehalose preconditioning increased autophagy flux. Trehalose preconditioning increased cardiomyocyte viability through the activation of autophagy in a DCD model, which could be a promising strategy for the prevention of cardiomyocyte damage in DCD transplantation.

**Keywords:** autophagy flux; donation after circulatory death; trehalose; cardiomyocyte

## 1. Introduction

Heart transplantation is the standard treatment for patients with end-stage heart failure that is refractory to medical treatment. A traditional heart transplantation is performed through donation after brain death (DBD), in which hearts are retrieved from heart-beating donors under life support measures. Despite the increasing number of heart transplantations worldwide, the shortage of donor hearts remains a significant problem [1]. Donation after circulatory death (DCD) can increase the number of donors and has been adopted in Australia and the UK. However, DCD is not accepted worldwide because of the high risk of complications and heart graft dysfunction. The rate of moderate or severe primary graft dysfunction is reportedly 22% in DCD compared with 10% in DBD [2].

The heart transplantation process in DCD differs from that in DBD because the donor hearts stop beating before retrieval, systolic blood pressure and systemic arterial oxy-gen saturation fall, and thus, a significant warm ischemic time (WIT) exists [3]. During WIT, there is a special period called the functional warm ischemic time (fWIT), during which the hearts are exposed to almost complete warm myocardial ischemia. After cardiac arrest, the heart is usually reperfused using extracorporeal perfusion machines before transplantation [4]. During DCD heart transplantation, warm ischemia and reperfusion contribute to organ damage known as ischemia-reperfusion (IR) injury. There is no established method for preventing IR injury during DCD heart transplantation.

Trehalose (Tre) is a naturally occurring disaccharide consisting of two glucose molecules linked together in a 1,1-glycosidic bond, and is widely known as an autophagy inducer. Autophagy is a fundamental cellular process that plays a crucial role in the maintenance of cellular homeostasis and the promotion of cell survival. Autophagy involves three main processes: the formation of autophagosomes, the fusion of autophagosomes with autolysosomes, and degradation. During this process, the damaged proteins and organelles are degraded and released into the cytoplasm for reuse. The overall autophagic activity is referred to as "autophagy flux". In a renal IR model, treatment with Tre mitigated IR-induced injury by promoting autophagosome and lysosome fusion, indicative of activated autophagy [5]. Furthermore, in our prior investigation employing a Langendorff rat heart model, pretreatment with Tre demonstrated enhanced cardiac function following IR injury, coupled with elevated levels of autophagy markers [6]. However, there have been no studies on the effects of Tre on autophagic flux in DCD.

Therefore, we examined the effects of Tre on cell viability in a DCD rat cardio-myocyte model. We hypothesized that, in the DCD model, Tre treatment would improve cell viability through the activation of autophagy.

## 2. Materials and Methods

### 2.1. Cell Culture

H9C2 cells (a rat cardiomyoblast cell line) were purchased from KAC Co., Ltd. (Tokyo, Japan) and cultured in Dulbecco's modified Eagle's medium (DMEM) with a high level of glucose (Gibco, Carlsbad, MA, USA) supplemented with 10% fetal bovine serum (Gibco) and 1% penicillin–streptomycin (10,000 U/mL, Gibco) according to the manufacturer's instructions.

### 2.2. Assessment of Cellular Toxicity for Trehalose and Chloroquine

The toxicity of trehalose (Tre, Hayashibara Co., Ltd., Okayama, Japan) and chloroquine (CQ, FUJIFILM Wako Pure Chemical Corporation, Osaka, Japan) was examined in advance. H9C2 cells were cultured overnight in 96-well plates. The culture medium was replaced by DMEM (Gibco) with different concentrations of the substances (Tre: 0, 10, 50, 100, 200, 1000 mM; CQ: 0, 10, 50, 100, 500, 1000, 2000, 5000 μM). After 4 h of exposure, Tre or CQ was replaced with normal cell culture medium containing 10% Cell Counting Kit-8 (CCK-8, Dojindo, Mashiki, Japan), and the cells were incubated for 1 h. Optical density (OD) values for all wells were obtained at 450 nm using a spectrophotometric system (Bio-Rad, Hercules, CA, USA). Cell viability (%) was calculated by the OD value of each treatment group divided by those of the 0 concentration groups. The concentrations at which the cellular viability significantly decreased were considered toxic, and the highest concentration that did not exhibit toxicity was used in further experiments.

### 2.3. Protocol of a Cellular DCD Model and Trehalose Treatments

A cellular DCD model was established to simulate a clinical situation. Figure 1 illustrates the protocol of the DCD model. At first, cells were cultured in 37 °C DMEM (Gibco) for 30 min at an $O_2$ level of 20%. The $O_2$ level was maintained at 1% for 60 min (95% $N_2$ and 5% $CO_2$, simulated WIT). To mimic functional warm ischemic conditions (fWIT), the culture medium was substituted with Dulbecco's Modified Eagle Medium (DMEM, Gibco) in PBS (Gibco) during the final 30 min to eliminate nutritional support from DMEM. Next, PBS (Gibco) was removed and cold 100 μL cardioplegic solution, St. Thomas no 2. (ST2, 4 °C), was added to each well. The composition of ST2 is 120 mM NaCl (FUJIFILM Wako Pure Chemical Corporation), 16 mM KCl (Sigma-Aldrich, St. Louis, MO, USA), 16 mM $MgCl_2$ (Sigma-Aldrich), 1.2 mM $CaCl_2$ (FUJIFILM Wako Pure Chemical Corporation), 10 mM $NaHCO_3$ (Sigma-Aldrich). Then, cells were incubated in a cold hypoxia environment (1% $O_2$ and 4 °C) for 1 h. Finally, the ST2 was replaced by a warmed DMEM (Gibco, 37 °C) and the cells were cultured for 1 h in a normal environment (20% $O_2$ and 37 °C), simulating reperfusion.

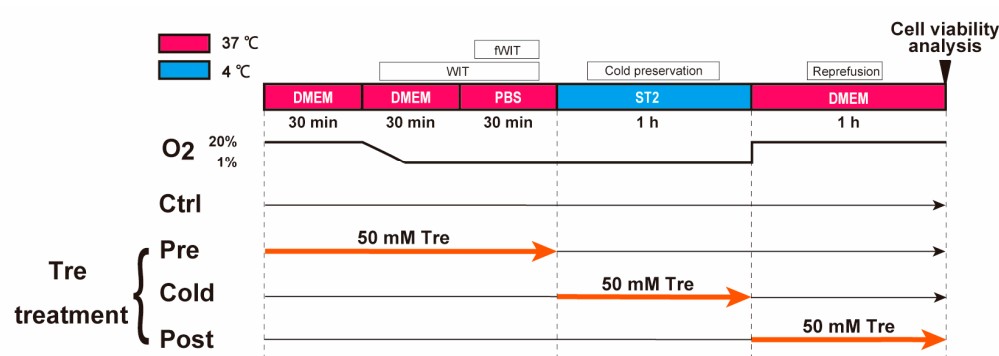

**Figure 1.** Protocol of a cell model to simulate heart transplantation from donation after circulatory death. Cells were cultured in a warm hypoxia environment (1% $O_2$ and 37 °C) for 60 min (WIT). The culture medium was switched from DMEM (Gibco) to PBS (Gibco) in the last 30 min to simulate fWIT. Next, the PBS was removed, and cold 100 µL cardioplegia solution, St. Thomas solution no 2. (ST2, 4 °C), was added to each well under a cold hypoxia environment (1% $O_2$ and 4 °C) for 1 h. Finally, the ST2 solution was replaced with a warmed DMEM culture medium (37 °C) and the cells were cultured for 1 h in a normal environment (20% $O_2$ and 37 °C), simulating reperfusion. Trehalose (Tre, 50 mM) was added at three different periods: (1) from preconditioning to fWIT (Pre), (2) cold preservation (Cold), and (3) reperfusion (Post). Ctrl group was not treated with Tre. Ctrl: control group in the DCD model; DMEM, Dulbecco's Modified Eagle Medium; fWIT, functional warm ischemic time; PBS, phosphate-buffered saline; WIT, warm ischemic time.

To determine the best time point for Tre treatment, 50 mM Tre was used at three different time points in the DCD model: (1) from preconditioning to fWIT (Pre), (2) cold preservation (Cold), and (3) reperfusion (Post).

### 2.4. Validation of Oxygen Concentrations and pH in Different Solutions

DMEM (Gibco) and PBS (Gibco) were exposed to an environment of 1% $O_2$ at 37 °C for 30 min to achieve hypoxic conditions during WIT and fWIT. To mimic the conditions of cold preservation, PBS was placed in the same hypoxic environment for 30 min followed by 1 h at 4 °C. The oxygen concentrations of different solutions were measured using a Strathkelvin 782 2-Channel Oxygen System (Motherwell, Scotland, UK). Briefly, after calibration, the electrode was inserted into the different solutions mentioned above and the oxygen concentrations of the solutions before and after exposure to the hypoxic environment were recorded. As indicated in Table 1, the oxygen concentrations decreased by more than 50% in all solutions after being exposed to the hypoxic environment for 30 min, with or without an additional at 4 °C.

**Table 1.** Oxygen concentrations of different solutions before and after hypoxia.

| Solutions | Oxygen Concentrations | | Decrease Rate (%) |
|---|---|---|---|
| | Before Hypoxia | After Hypoxia | |
| DMEM (37 °C) | 1.61 µL/mL | 0.68 µL/mL | 58 |
| PBS (37 °C) | 6.20 mL/L | 2.30 mL/L | 63 |
| PBS (4 °C) | 14.60 mg/L | 7.18 mg/L | 51 |

The pH levels of different solutions before and after WIT, fWIT, and cold preservation were also measured using a pH meter (DKK-TOA Corporation, Tokyo, Japan). Following calibration, the electrode was immersed into the respective solutions, and the pH values were recorded both before and after exposure to hypoxic conditions. Additionally, the pH of DMEM (Gibco) was measured before and after 30 min and 1 h of culture as a reference. Table 2 illustrates that pH levels decreased to a greater extent under ischemic conditions compared to the normal group, which suggested that the model simulated the ischemia condition, rather than solely hypoxia, during WIT and fWIT.

**Table 2.** pH of different solutions.

| Solutions | pH | | Decrease |
|---|---|---|---|
| | Before | After | |
| DMEM (37 °C, 30 min hypoxia) | 8.18 | 7.80 | 0.38 |
| PBS (37 °C, 30 min hypoxia) | 7.41 | 6.74 | 0.67 |
| ST2 (4 °C, 1 h hypoxia) | 7.89 | 6.92 | 0.97 |
| DMEM (37 °C, 30 min normoxia) | 8.20 | 8.03 | 0.17 |
| DMEM (37 °C, 1 h normoxia) | 8.35 | 8.01 | 0.34 |

### 2.5. Assessment of Cell Viability in the DCD Model

To examine the protective effects of Tre, cell viability in the DCD model was assessed using CCK-8 (Dojindo) and Propidium Iodide (PI)/Hoechst 33342 (Dojindo) staining. Cell viability was measured using the CCK-8 assay after reperfusion and calculated as previously described. For PI/Hoechst 33342 (Dojindo) staining, after 1 h reperfusion, cells were washed with PBS (Gibco) once and incubated with 100 µL PBS (Gibco) containing 0.67 µg/mL PI (Dojindo) and Hoechst 33342 (Dojindo) dye in each well for 30 min according to the manufacturer's instructions. PI (Dojindo) was used for dead cell staining, which is shown in red, and Hoechst (Dojindo) was used for all cell staining, which is shown in blue. Images were obtained using the APX100 Digital Imaging System (EVIDENT CORPORATION, Tokyo, Japan), and cell numbers were counted using ImageJ software Version 1.54 (U. S. National Institutes of Health, Bethesda, MD, USA). The percentage of live cells was calculated as the live cell number divided by the total cell number. To exclude the effect of osmolarity, we also used mannitol (50 mM, FUJIFILM Wako Pure Chemical Corporation) and sucrose (50 mM, FUJIFILM Wako Pure Chemical Corporation) as negative controls during the preconditioning and ischemia phases. Subsequently, we tested the cell viability after reperfusion by CCK-8 (Dojindo).

### 2.6. Assessment of Cell Viability after Using Chloroquine, an Autophagy Inhibitor

Because Tre is an autophagy activator, we attempted to determine the role of autophagy in the protective effects of Tre. CQ is a well-known inhibitor of autophagy. In this experiment, we used CQ at a maximum concentration of 50 µM without toxicity. Figure 2 shows the protocol for CQ administration in the control and Tre groups. Cell viability analysis using the CCK-8 assay was performed after reperfusion as described previously.

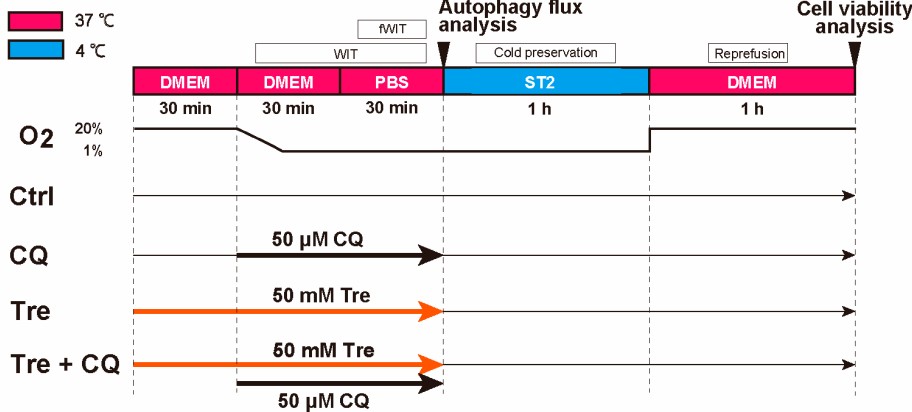

**Figure 2.** Protocol of autophagy evaluation under trehalose (Tre) preconditioning. Tre was used from preconditioning to functional warm ischemic time (fWIT) in Tre and Tre + chloroquine (CQ, an autophagy inhibitor) groups. CQ was used from warm ischemic time (WIT) to fWIT in CQ and Tre + CQ groups. The concentrations of Tre and CQ were 50 mM and 50 µM, respectively. Protein extraction for autophagy flux was performed just after fWIT. Cell viability was evaluated by Cell Counting Kit-8 (CCK-8) incubation for 1 h after Reperfusion. Ctrl: control group in the DCD model.

### 2.7. Analysis of Autophagy Flux

The assessment of autophagic flux is essential for evaluating autophagic activity. Autophagic flux can be assessed based on an increase in autophagosomes after the application of lysosome inhibitors. CQ is one of the most commonly used lysosomal inhibitors and blocks the fusion of autophagosomes with autolysosomes. The number of autophagosomes was estimated by measuring the expression of microtubule-associated protein 1A/1B light chain 3 B (LC3)-II using western blotting. The differences in the expression of LC3-II between samples in the presence and absence of CQ represent the number of autophagosomes delivered to lysosomes for degradation (i.e., autophagic flux) [7].

For protein expression analysis following Tre treatment, proteins were extracted from the four groups after fWIT (Figure 2). After adding lysis buffer (FUJIFILM Wako Pure Chemical Corporation) containing 1% proteinase inhibitor (FUJIFILM Wako Pure Chemical Corporation), the total protein expression was assessed using the Bradford method. A semi-dry Western blot apparatus (Mini-PROTEAN Tetra Cell; Bio-Rad) was used to detect the conversion from LC3-I (cytosolic form) to LC3-II (membrane-bound lipidated form). After sodium dodecyl sulfate-polyacrylamide gel electrophoresis (12% Mini-PROTEANTGX; Bio-Rad), proteins were blotted onto polyvinylidene difluoride membranes and incubated with primary (Anti-LC3B; 1:500, Abcam, Cambridge, UK) and secondary antibodies (Anti-rabbit IgG; 1:2000, Cell Signaling Technology, Danvers, MA, USA). The bands were semi-quantified by chemiluminescence using JustTLC (Sweday, Sodra Sandby, Sweden). The membranes were then stained with an amido black solution to normalize the band intensity.

### 2.8. Statistical Analysis

All data are presented as mean $\pm$ SEM. Student's *t*-test was used for two-group comparisons of means, while one-way analysis of variance (ANOVA) was used for comparisons between more than two groups, followed by Dunnett's test or the Mann–Whitney U test for pairwise comparisons. $p < 0.05$ was considered statistically significant. GraphPad Prism, Version 9 (GraphPad Software, Boston, MA, USA) was used for all the analyses and figures. The effect size (Cohen's d) was calculated, and a value higher than 2.0 was considered large.

## 3. Results

### 3.1. Cellular Toxicity of Trehalose

As shown in Figure 3, cell viability did not show significant differences when treated with Tre at concentrations lower than 100 mM and was maintained at over 98%. The cell viability decreased with increasing concentrations (100 mM, 85%, $p < 0.0001$; 200 mM, 89%, $p = 0.002$; 1000 mM, 26%, $p < 0.0001$). The maximum concentration of Tre that did not cause cellular toxicity (50 mM) was used in further experiments.

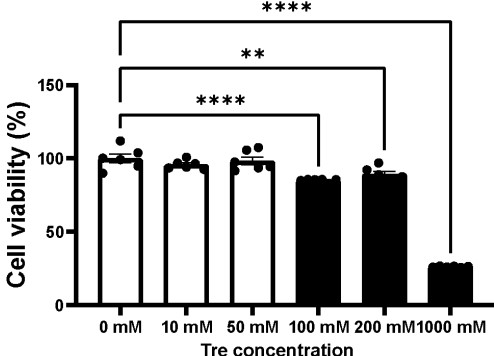

**Figure 3.** Cellular toxicity assessment of trehalose (Tre). Cell viability was evaluated after 4 h treatment with Tre of different concentrations using Cell Counting Kit-8 (n = 6). The bar graphs indicate the means $\pm$ SEM. One-way analysis of variance (ANOVA) was used for comparisons followed by the Dunnett's test. ** $p < 0.01$, and **** $p < 0.0001$.

### 3.2. Effects of Trehalose Treatments on Cell Viability in the DCD Model

To investigate the potential effects of Tre in the DCD model, Tre was used at different time points, from preconditioning to fWIT, cold preservation, or reperfusion. Figure 4 shows the cell viability results for Tre at different time points. When used in preconditioning, the Tre group showed a higher amount of cell viability after reperfusion than the control group, with a considerable effect size (Figure 4A, 69% vs. 55%, $p < 0.0001$, effect size $d = 5.79$). When used in cold preservation, the Tre group also showed a higher amount of cell viability than the control group (Figure 4B, 53% vs. 49%, $p = 0.04$, effect size $d = 1.12$). In contrast, when used in reperfusion, the Tre group showed a lower amount of cell viability than the control group (Figure 4C, 73% vs. 77%, $p = 0.010$, effect size $d = 1.42$). Because Tre treatment during preconditioning showed the most significant effect on cell viability, we focused on Tre preconditioning in subsequent experiments. We further examined cell viability following the application of mannitol and sucrose during preconditioning and ischemia to evaluate the influence of osmolarity. No increase in cell viability was observed after treatment with mannitol and sucrose (Figure S1).

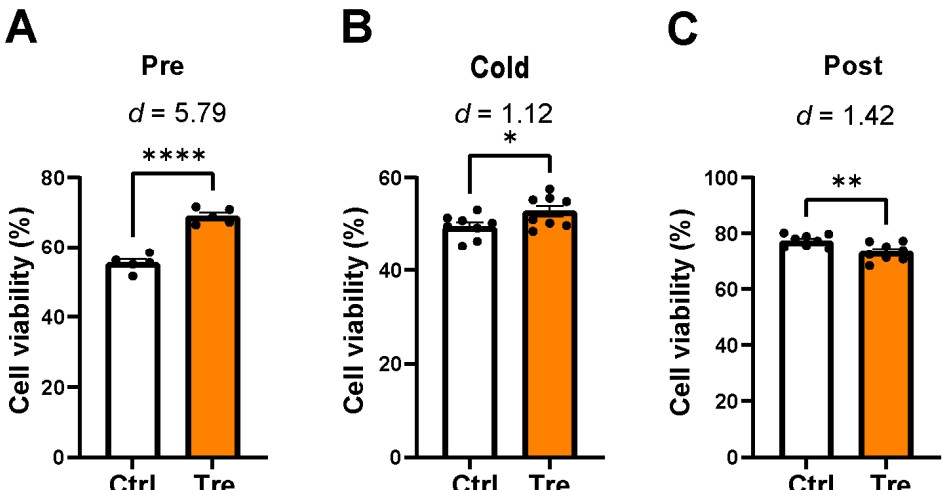

**Figure 4.** Cell viability after trehalose (Tre) treatments in a cell model of heart transplantation from donation after circulatory death. Ctrl: control group in the DCD model. Tre (50 mM) was used at three different time points: from preconditioning to functional warm ischemic time (**A**, Pre), cold preservation (**B**, Cold), and reperfusion (**C**, Post). Cell viability was measured after reperfusion. The bar graphs indicate the means ± SEM. n = 5 in the preconditioning group, n = 8 in the cold preservation and postconditioning groups. Student's $t$-test was used for two-group comparisons of means. * $p < 0.05$, ** $p < 0.01$, and **** $p < 0.0001$. The effect size was calculated as Cohen's $d$.

To confirm the cell viability results using CCK-8, we performed PI/Hoechst staining, which showed similar results. Figure 5A shows representative fluorescence images. Fewer blue-stained cells were observed in the control group in the DCD model than in the no-ischemia group, suggesting cell loss in the DCD model. Because we used PBS only once to reduce cell loss by washing, most of the lost cells were considered dead. The Tre-treated group showed a higher cell density than the control group, suggesting decreased cell loss. Consistent with the fluorescence images, the Tre treatment group showed an increased tendency of calculated live cell percentage compared to the control group (Figure 5B, 78% vs. 64%, $p = 0.055$, effect size $d = 0.81$).

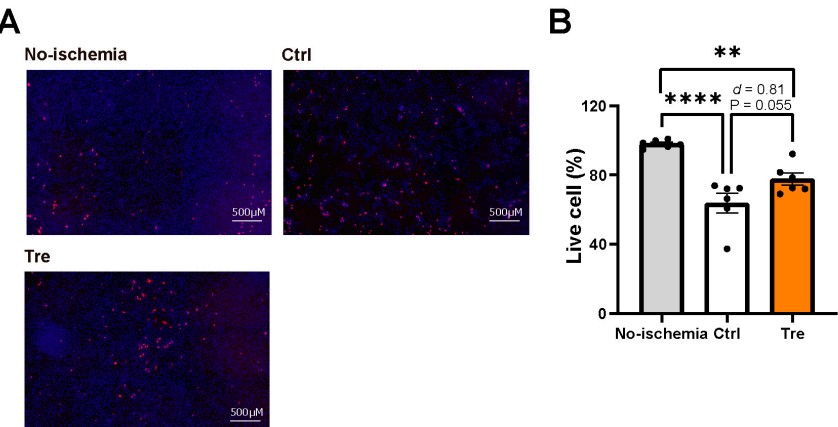

**Figure 5.** Cell viability evaluated by Propidium Iodide (PI)/Hoechst staining after trehalose (Tre) preconditioning. No-ischemia: normal cells without undergoing ischemia-reperfusion. Ctrl: control group in the DCD model. (**A**) Representative images of PI/Hoechst staining after 1 h reperfusion. PI was used for dead cell staining, which showed in red, and Hoechst for all cell staining, which showed in blue. The images were obtained using an immunofluorescence microscopy system. (**B**) Live cell percentage calculated as live cell numbers divided by total cell numbers (n = 6). The bar graphs indicate the means ± SEM. One-way analysis of variance (ANOVA) was used for comparisons followed by the Mann–Whitney U test. ** $p < 0.01$ and **** $p < 0.0001$. The effect size was calculated as Cohen's *d*.

### 3.3. Cell Viability after Trehalose Using Autophagy Inhibitor, Chloroquine

Tre is a well-known autophagy activator. It is highly likely that autophagy is related to the protective effect of Tre in our DCD model. Therefore, we used the autophagy inhibitor, CQ, to examine the role of autophagy in Tre preconditioning.

First, we assessed the toxicity of CQ to determine its concentration. As shown in Figure 6A, higher concentrations of more than 50 μM lead to significant decreases in cell viability compared to that of the 0 μM group (100 μM, 93%, $p = 0.016$; 500 μM, 84%, $p < 0.0001$; 1000 μM, 37%, $p < 0.0001$; 2000 μM, 27%, $p < 0.0001$; 5000 μM, 28%, $p < 0.0001$), which suggested that the concentrations more than 50 μM were toxic. We used CQ at a concentration of 50 μM in our following experiments based on these results.

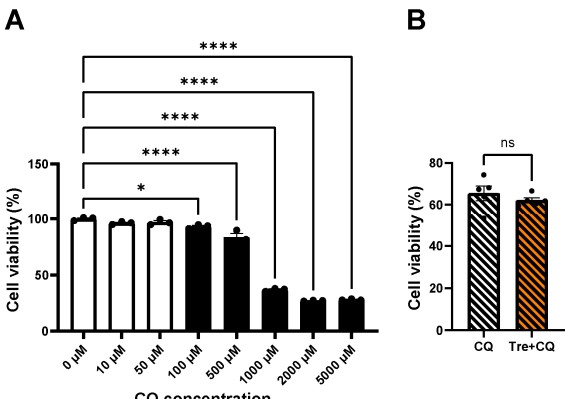

**Figure 6.** Chloroquine (CQ) toxicity assessment and cell viability in trehalose (Tre) preconditioning using CQ. (**A**) CQ toxicity assessment. Cell viability was evaluated after 4 h treatment with CQ of different concentrations using Cell Counting Kit-8 (n = 3). One-way analysis of variance (ANOVA) was used for comparisons followed by the Dunnett's test. * $p < 0.05$, and **** $p < 0.0001$. (**B**) Cell viability was assessed between CQ and Tre + CQ group after reperfusion in a cell model of heart transplantation from donation after circulatory death (n = 5). The concentrations of Tre and CQ were 50 mM and 50 μM, respectively. Student's *t*-test was used for two-group comparisons of means. ns indicates no significant difference. The bar graphs indicate the means ± SEM.

Figure 6B shows cell viability under the CQ treatment. There was no significant difference in cell viability between the CQ and Tre + CQ groups (65% vs. 62%, $p = 0.39$). This suggests that the protective effects of Tre were canceled out by CQ and that autophagy could play an important role in these effects.

### 3.4. Autophagy Flux in Trehalose Preconditioning

To further understand the autophagic flux after Tre preconditioning, we used the lysosome inhibitor CQ during the WIT and fWIT periods in the DCD model. Figure 7A shows no difference in LC3-II expression between the control and CQ groups (1.00 vs. 1.04 arbitrary unit, $p = 0.85$, effect size $d = 0.2$), suggesting no significant autophagy flux in the control group. On the other hand, compared to the Tre group, the Tre + CQ group showed a higher LC3-II expression (Figure 7B, 1.00 vs. 2.15 arbitrary unit, $p = 0.025$, effect size $d = 2.7$), suggesting that there was more autophagy flux after Tre preconditioning compared with the control.

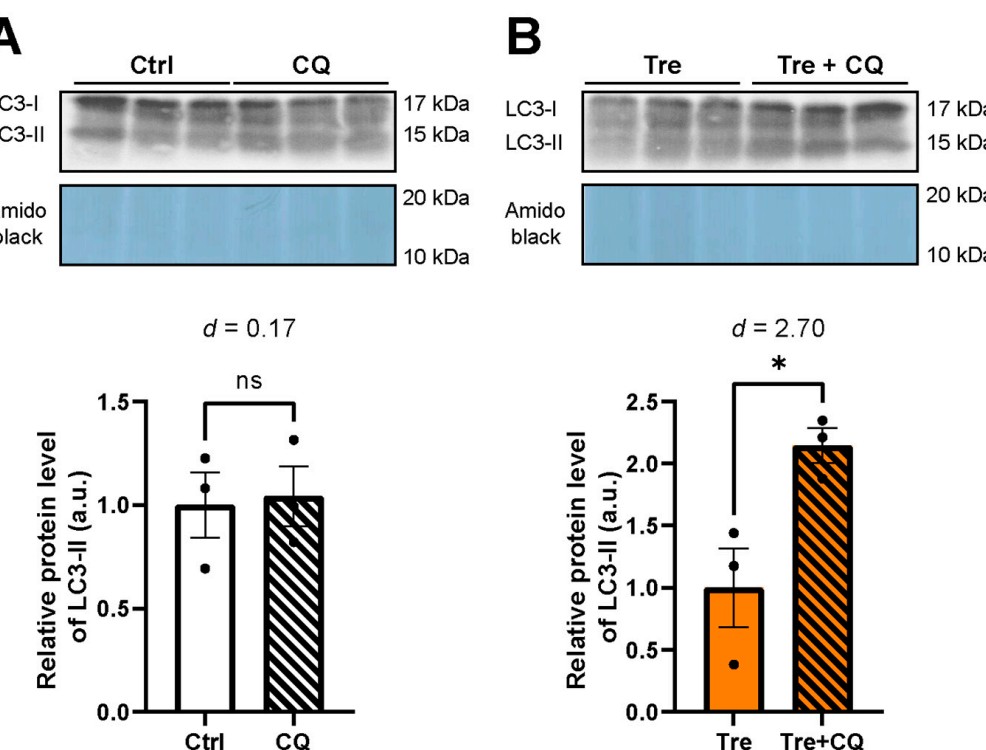

**Figure 7.** Assessment of autophagy flux using chloroquine (CQ). The expression of microtubule-associated proteins 1A/1B light chain 3B (LC3) was examined by Western blotting. The band intensity of LC3-II was normalized by amido black staining of the membranes. Ctrl: control group in DCD model. (**A**) Autophagy flux in the Ctrl group (n = 3). ns indicates no significant difference. (**B**) Autophagy flux in trehalose (Tre) group (n = 3). * $p < 0.05$. The effect size was calculated as Cohen's $d$. Student's $t$-test was used for two-group comparisons of means. The bar graphs indicate the means ± SEM.

### 4. Discussion

In this study, we demonstrated that Tre preconditioning increased H9C2 cell viability and that autophagy activation contributed to the protective effects of Tre in a DCD model. Tre preconditioning activated autophagy, and the effects of Tre were canceled out by an autophagy inhibitor. Thus, we suggest that the protective effect of Tre is related to autophagy activation.

Preconditioning is an intervention aimed at preparing and protecting the tissues and organs from subsequent stress or injury. Preconditioning exposes tissues or organs to mild or transient stress that triggers adaptive responses and activates protective mechanisms. These mechanisms protect tissues and organs against severe stress or injury, thereby

reducing damage [8,9]. There are several types of preconditioning, including ischemic, pharmacological, and remote preconditioning. Pharmacological preconditioning is considered the easiest method to apply, and we found that preconditioning with Tre was protective in this study. Autophagy is associated with several types of pharmacological preconditioning. A study using resveratrol preconditioning in a mouse myocardial IR model revealed that preconditioning protected the heart by activating autophagy [10]. Another study that used sevoflurane preconditioning in a mouse IR model showed that infarct size and cardiac damage were alleviated by autophagy [11].

Tre is a well-known activator of autophagy that acts via several pathways. Transcription factor EB (TFEB), a master gene for lysosomal biogenesis, is one of the ways that Tre activates autophagy. In a mouse model of chronic ischemic cardiomyopathy, Tre application led to increased nuclear localization of TFEB and induced autophagy in the hearts [12]. In addition, Tre induces autophagy via the p38-MAPK pathway in a mouse model of insulin resistance [13]. These pathways may explain the activation of autophagy during Tre preconditioning. However, studies on the mechanism of autophagy activation by Tre preconditioning are limited and further evaluation is necessary [6].

In our study, Tre activated autophagy to increase cell viability; however, other mechanisms have also been reported for the therapeutic effects of Tre. Several studies have reported antioxidant effects of Tre. Forkhead box O1 (FOXO1) is a common protein. FOXO1 is highly enhanced by Tre [14], and can regulate the expression of superoxide dismutases and catalase [15]. Therefore, FOXO1 may play an essential role in the antioxidant function of Tre. Additionally, Tre is involved in the inhibition of ferroptosis. Ferroptosis is the primary type of cell death that occurs during exposure to cold and IR. Excessive oxidative stress including lipid peroxidation is the primary cause of ferroptosis [16]. In a rat model of myocardial IR, ferroptosis occurred, leading to worse heart function [17]. Tre inhibited ferroptosis via the NRF2/HO-1 pathway in a mouse model of spinal cord injury with reduced lipid peroxide production [18]. Taken together, these results suggest that Tre may play a role in alleviating oxidative stress in our DCD model.

In our study, we utilized a concentration of 50 mM Tre, and the CCK-8 assay results revealed no significant decrease in cell viability following a 4 h treatment. While our concentration determination experiment did not assess subtle cellular changes, which may warrant further investigation. Trehalose is known as a cell membrane protector, with established roles in cell cryoprotection [19]. Moreover, Trehalose has been utilized in ET-Kyto solution for long-term lung preservation at a concentration of 120 mM [20]. Therefore, the potential cardioprotective effects of Trehalose, even at a concentration of 50 mM, remain promising.

This study had some limitations. First, we did not examine the mechanism underlying the pathways through which trehalose modulates autophagy or the effects of different glucose concentrations in the medium in this study, which require further investigation. Second, it is also a limitation of our study that we did not use differentiated H9C2 cells, although there are a few studies suggesting that differentiated cells are closer to mature cardiomyocytes. Third, the effects of Tre may need to be verified using animal models in the future. Lastly, CQ is not a selective autophagy inhibitor. Therefore, we cannot exclude other mechanisms that may affect Tre preconditioning.

In summary, Tre increased cardiomyocyte viability in the DCD model, especially during preconditioning. Autophagy inhibition negates the protective effects of Tre preconditioning. Autophagy activation is a potential mechanism by which Tre preconditioning contributes to the improved cell viability in this DCD model.

**Supplementary Materials:** The following supporting information can be downloaded at: https://www.mdpi.com/article/10.3390/cimb46040210/s1, Figure S1: assessment of osmolarity effect.

**Author Contributions:** Conceptualization, J.G. and Y.S.; methodology, J.G. and Y.S.; validation, S.W.; formal analysis, J.G.; investigation, J.G. and Y.S.; writing—original draft preparation, J.G.; writing—review and editing, Y.S. and S.W.; visualization, S.W.; supervision, S.W. All authors have read and agreed to the published version of the manuscript.

**Funding:** This research received no external funding.

**Institutional Review Board Statement:** Not applicable.

**Informed Consent Statement:** Not applicable.

**Data Availability Statement:** Data are available on request to the authors.

**Conflicts of Interest:** The authors declare no conflicts of interest.

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
