# Peer review of "Effects of Trehalose Preconditioning on H9C2 Cell Viability and Autophagy Activation in a Model of Donation after Circulatory Death for Heart Transplantation"

_cimb, doi:10.3390/cimb46040210_

Round 1
Reviewer 1 Report
Comments and Suggestions for Authors
In this study the authors used H9C2 cells as a model to test pharmaca (here trehalose) for DCD preservation.
Major comments:
I have two main points. The authors should replace the term ischemia by hypoxia and reperfusion by reoxygenation as this is what they did. During ischemia in tissues you can observe acidosis that is not covered by the current protocol.
Figure 7: The so-called amido black staining are unacceptable in quality and grade of information. Please add a suitable loading control such as GAPDH, Actin or others. Nay information about the WB method and antibodies is missing.
Minor points
Please give a reference to the composition of ST2 solution.
Author Response
First of all, we sincerely thank the Reviewer #1 for all the valuable comments to improve our manuscript.
- Comments to the Authors:
The authors should replace the term ischemia by hypoxia and reperfusion by reoxygenation as this is what they did. During ischemia in tissues you can observe acidosis that is not covered by the current protocol.
Author’s response:
We appreciate the comments from the Reviewer. As the reviewer suggested, we added monitoring of the pH levels in various media at different time points. We observed a decrease in pH during the ischemic period (from WIT to cold preservation) compared to cells cultured under normal conditions. Following fWIT and cold preservation, the pH decreased to 6.74 and 6.92, respectively. We have included the pH decrease data in section 2.4 "Validation of oxygen concentrations and pH in different solutions".
In our protocol, we not only adjusted the oxygen concentration to 1%, but also switched the culture medium from DMEM to PBS and ST2 solution, which are saline buffers, and then back to DMEM to mimic the conditions of ischemia and reperfusion. Therefore, we utilized the terms ischemia and reperfusion in our study. We hope these will align with your expectations.
(Materials and Methods, Page 3-4)
The pH levels of different solutions before and after WIT, fWIT and cold preservation were also measured by a pH meter (DKK-TOA CORPORATION, Tokyo, Japan). Following calibration, the electrode was immersed into the respective solutions, and the pH values were recorded both before and after exposure to hypoxic conditions. Additionally, the pH of DMEM (Gibco) was measured before and after 30 minutes and 1 hour of culture as a reference. Table 2 illustrates that pH levels decreased to a greater extent under ischemic conditions compared to the normal group.
Table 2. pH of different solutions
|
Solutions |
pH |
Decrease |
|
|
|
Before |
After |
|
||
|
DMEM (37 ℃, 30 min hypoxia) |
8.18 |
7.80 |
0.38 |
|
|
PBS (37 ℃, 30 min hypoxia) |
7.41 |
6.74 |
0.67 |
|
|
ST2 (4 ℃, 1 h hypoxia) |
7.89 |
6.92 |
0.97 |
|
|
DMEM (37 ℃, 30 min normoxia) |
8.20 |
8.03 |
0.17 |
|
|
DMEM (37 ℃, 1 h normoxia) |
8.35 |
8.01 |
0.34 |
|
- Comments to the Authors:
Figure 7: The so-called amido black staining are unacceptable in quality and grade of information. Please add a suitable loading control such as GAPDH, Actin or others. Any information about the WB method and antibodies is missing.
Author’s response
Thank you for your advice. We appreciate your suggestion regarding the addition of a loading control such as GAPDH or Actin. While GAPDH and actin are commonly utilized as loading controls, in our study, we encountered significant variations in GAPDH expression among different treatment groups, likely influenced by ischemia treatment. Consequently, we opted not to employ GAPDH as a loading control. Instead, we utilized amido black staining to visualize total protein on the membrane, a method supported by previous research (Georgina M Aldridge et al., J Neurosci Methods, 2008). We have revised Figure 7 to improve contrast and clarity.
Furthermore, we have provided details on the antibodies used in our study, specifically in section 2.7 "Analysis of autophagy flux". We trust that these adjustments may meet your expectations.
(Materials and Methods, Page 5)
After sodium dodecyl sulfate-polyacrylamide gel electrophoresis (12% Mini-PROTEANTGX; Bio-Rad), proteins were blotted onto polyvinylidene difluoride membranes and incubated with primary (Anti-LC3B; 1: 500, Abcam, Cambridge, UK) and secondary antibodies (Anti-rabbit IgG; 1:2000, Cell Signaling Technology, MA, USA).
- Comments to the Authors:
Please give a reference to the composition of ST2 solution.
Author’s response:
Thank you for your advice. We have revised this part according to the reviewer’s suggestion in page 2-3, line 93-96.
(Materials and Methods, Page 2-3)
The composition of ST2 is 120 mM NaCl (FUJIFILM Wako Pure Chemical Corporation), 16 mM KCl (Sigma-Aldrich, MO, USA), 16 mM MgCl2 (Sigma-Aldrich), 1.2 mM CaCl2 (FUJIFILM Wako Pure Chemical Corporation), 10 mM NaHCO3 (Sigma-Aldrich).
Reviewer 2 Report
Comments and Suggestions for Authors
Dear Editors,
The paper addresses a highly relevant area of research, given the critical shortage of donor hearts for transplantation and the associated complications from donation after circulatory death (DCD). This study provides insights into the potential of trehalose preconditioning as a strategy to enhance cell viability and reduce IR injury in the context of DCD heart transplantation. However, the transition from in vitro findings to clinical application requires careful consideration of the model's limitations, deeper mechanistic studies, and validation in more physiologically relevant models.
Major
- The use of Dulbecco’s Modified Eagle’s Medium (DMEM) supplemented with fetal bovine serum and penicillin-streptomycin is standard. However, the study does not discuss whether the glucose levels in the DMEM (high glucose variant) could impact cellular responses to ischemia and trehalose treatment. Glucose metabolism plays a critical role in ischemic conditions and high glucose is known to inhibit autophagy; thus, its concentration in the culture media could influence the study outcomes.
- What was the CO2 concentration during the DCD protocol? Did the pH of the culture media change?
- Trehalose alters the osmolarity of the culture medium. How might this affect the results? Is there any negative control that might have been used to account for the effects of the higher osmolarity? Sucrose for example?
- The assessment of trehalose cellular toxicity using only CCK8 is not enough. Further characterization is required. Other subtle cellular changes might have occurred at lower concentrations. Moreover, cellular toxicity should be evaluated at longer periods to check for long term effects.
- While the study implicates autophagy activation in the protective effects of trehalose, it does not deeply investigate the pathways through which trehalose modulates autophagy or other cell survival mechanisms. Further elucidation of these pathways would strengthen the study's conclusions. Furthermore, exploration into the mechanisms by which trehalose exerts its effects at specifically the preconditioning time would add depth to the findings.
- The PI photos for Ctrl and Tre in figure 5 seem similar.
- In figure 7, the results seem contradictory to the rest of the paper. How did LC3 expression increase under chloroquine, since the latter is an autophagy inhibitor? Besides, in panel B, LC3 expression is lower under trehalose as compared to control in panel A. How do the authors explain these discrepancies?
Minor
- The previous use of trehalose in ischemia-reperfusion should be more detailed in the introduction.
- The authors should further explain how replacing medium with DMEM in PBS simulates fWIT.
Comments on the Quality of English LanguageMinor editing is required.
Author Response
First of all, we sincerely thank the Reviewer #2 for all the valuable comments to improve our manuscript.
- Comments to the Authors
The study does not discuss whether the glucose levels in the DMEM (high glucose variant) could impact cellular responses to ischemia and trehalose treatment. Glucose metabolism plays a critical role in ischemic conditions and high glucose is known to inhibit autophagy; thus, its concentration in the culture media could influence the study outcomes.
Author’s response:
Thank you for your advice. We agree that high glucose is known to inhibit autophagy. however, trehalose-induced autophagy activation was not inhibited by high glucose concentrations in our study. We have revised the limitation stated below.
(Discussion, Page 10, Line 327-329)
First, we did not examine the mechanism underlying the pathways that trehalose modulates autophagy or the effects of different glucose concentrations in the medium in this study, which requires further investigation.
- Comments to the Authors:
What was the CO2 concentration during the DCD protocol? Did the pH of the culture media change?
Author’s response:
Thank you for your insightful question. In our protocol, the CO2 concentration remains at 5%, consistent with a normal culture environment, while N2 is utilized to achieve the 1% O2 condition. Furthermore, we have revised the protocol, specifically on page 2, lines 88-89. Additionally, we have added measurements of pH in the culture media before and after hypoxia, and these details have been incorporated into section 2.4 "Validation of oxygen concentrations and pH in different solutions". We hope these adjustments will meet your expectations.
(Materials and Methods, Page 2, Line 88)
The O2 level was maintained at 1% for 60 min (95% N2 and 5% CO2, simulated WIT).
(Materials and Methods, Page 3-4, Line 129-136)
The pH levels of different solutions before and after WIT, fWIT and cold preservation were also measured by a pH meter (DKK-TOA CORPORATION, Tokyo, Japan). Following calibration, the electrode was immersed into the respective solutions, and the pH values were recorded both before and after exposure to hypoxic conditions. Additionally, the pH of DMEM (Gibco) was measured before and after 30 minutes and 1 hour of culture as a reference. Table 2 illustrates that pH levels decreased to a greater extent under ischemic conditions compared to the normal group.
Table 2. pH of different solutions
|
Solutions |
pH |
Decrease |
|
|
|
Before |
After |
|
||
|
DMEM (37 ℃, 30 min hypoxia) |
8.18 |
7.80 |
0.38 |
|
|
PBS (37 ℃, 30 min hypoxia) |
7.41 |
6.74 |
0.67 |
|
|
ST2 (4 ℃, 1 h hypoxia) |
7.89 |
6.92 |
0.97 |
|
|
DMEM (37 ℃, 30 min normoxia) |
8.20 |
8.03 |
0.17 |
|
|
DMEM (37 ℃, 1 h normoxia) |
8.35 |
8.01 |
0.34 |
|
- Comments to the Authors:
Trehalose alters the osmolarity of the culture medium. How might this affect the results? Is there any negative control that might have been used to account for the effects of the higher osmolarity? Sucrose for example?
Author’s response:
We appreciate the comments provided by the Reviewer. During the preconditioning and ischemia period, we added experiments using 50-mM mannitol or 50-mM sucrose concurrently to assess the impact of osmolarity. According to the CCK-8 results, the cell viability following treatment with mannitol and sucrose was comparable to that of the DCD control group (cell viability: DCD control: 73%, Sucrose: 68%, Mannitol: 67%), indicating no effect from increased osmolarity in this protocol.
- Comments to the Authors:
The assessment of trehalose cellular toxicity using only CCK8 is not enough. Further characterization is required. Other subtle cellular changes might have occurred at lower concentrations. Moreover, cellular toxicity should be evaluated at longer periods to check for long term effects.
Author’s response:
Thank you for your advice. We appreciate your concern regarding the assessment of trehalose cellular toxicity. We understand that trehalose, even at current or lower concentrations, could potentially induce subtle cellular changes beyond the scope of our initial evaluation using CCK8. However, in our experimental protocol, we limited 50-mM trehalose exposure to 90 minutes, with no trehalose present in subsequent solutions. We believe the timeframe of 4 h for toxicity was sufficient to evaluate short-term cell viability changes. We appreciate your consideration and will consider your suggestions for future studies to understand the effect of long-term trehalose treatment. Besides, 100-mM trehalose has already been used in long-term preservation solutions for the lung (ET-Kyoto solution).
- Comments to the Authors:
While the study implicates autophagy activation in the protective effects of trehalose, it does not deeply investigate the pathways through which trehalose modulates autophagy or other cell survival mechanisms.
Author’s response:
Thank you for your valuable comment. We appreciate your suggestion regarding the need for a deeper investigation into the pathways by which trehalose modulates autophagy and other cell survival mechanisms. We will address this gap in future research. Additionally, we have revised the discussion section (page 10, lines 327-329) accordingly. We hope these adjustments will meet your expectations.
(Discussion, Page 10, Line 327-329)
First, we did not examine the mechanism underlying the pathways that trehalose modulates autophagy or the effects of different glucose concentrations in the medium in this study, which requires further investigation.
- Comments to the Authors:
The PI photos for Ctrl and Tre in figure 5 seem similar.
Author’s response:
We appreciate the reviewer’s comment. While the images may appear similar at first glance, there are differences between the Ctrl and Tre groups in Figure 5. It becomes apparent that the control group exhibits significant cell loss with a higher number of dead cells, whereas the trehalose group shows limited cell loss and fewer instances of cell death.
The quality of the figures provided for review may have been affected by document compression, which could potentially impact their clarity. The attached figures may maintain better quality, facilitating clearer differences between the two groups.
- Comments to the Authors:
In figure 7, the results seem contradictory to the rest of the paper. How did LC3 expression increase under chloroquine, since the latter is an autophagy inhibitor? Besides, in panel B, LC3 expression is lower under trehalose as compared to control in panel A. How do the authors explain these discrepancies?
Author’s response:
Thank you for raising this question. Autophagy is a dynamic process involving the degradation of autophagosomes by lysosomes. While chloroquine is commonly known as an autophagy inhibitor, it primarily inhibits lysosomes rather than upstream autophagy processes. Consequently, when autophagy is activated, there is an accumulation of autophagosomes within cells when chloroquine is used, leading to an apparent increase in autophagosome marker, LC3 II expression.
Moreover, in instances of excessive autophagy activation, protein levels can decrease due to heightened degradation of LC3 II. Therefore, the LC3 II expression is lower in the trehalose group compared to the control group, which may suggest the activation of autophagy. Also, solely observing LC3 II expression may not provide a clear indication of autophagy flux. To address this, we used chloroquine as a lysosome inhibitor to validate the autophagy flux.
- Comments to the Authors:
The previous use of trehalose in ischemia-reperfusion should be more detailed in the introduction.
Author’s response:
We have revised the related part in the introduction (page 2, line 55-59).
(Introduction, Page 2, Line 55-59)
In a renal IR model, treatment with Tre mitigated IR-induced injury by promoting autophagosome and lysosome fusion, indicative of activated autophagy [5]. Furthermore, in our prior investigation employing a Langendorff rat heart model, pretreatment with Tre demonstrated enhanced cardiac function following IR injury, coupled with elevated levels of autophagy markers [6].
- Comments to the Authors:
The authors should further explain how replacing medium with DMEM in PBS simulates fWIT.
Author’s response:
Thank you for your comment. During fWIT, there is virtually almost no blood flow and thus no nutrients or oxygen present in the tissue. We mimic the situation by using PBS in a hypoxic environment. We have revised the related part in the methods (page 2, line 88-92).
As we mentioned above, we detected greater acidosis when using PBS compared to DMEM.
(Materials and Methods, Page 2, line 88-92)
The O2 level was maintained at 1% for 60 min (95% N2 and 5% CO2, simulated WIT). To mimic functional warm ischemic conditions (fWIT), the culture medium was substituted with Dulbecco's Modified Eagle Medium (DMEM, Gibco) in PBS (Gibco) during the final 30 minutes to eliminate nutritional support from DMEM.
Comments on the Quality of English Language
Minor editing is required.
Author’s response:
We appreciate the Reviewer’s comments. We modified sentences throughout the manuscript with native speaker’s support.
Round 2
Reviewer 2 Report
Comments and Suggestions for Authors
Dear Editors,
The authors have adequately addressed the concerns. The methods and results are now exhaustively described to allow further replication.
It is still recommended that the authors include the mannitol results, the trehalose toxicity (response 4) discussion, and the LC3 expression explanation with appropriate references in the paper. They should also discuss the pH results and how they might impact the study's outcomes.
Kind Regards
Author Response
First of all, we sincerely thank the Reviewer #2 for all the valuable comments to improve our manuscript
Comments to the Authors:
It is still recommended that the authors include the mannitol results, the trehalose toxicity (response 4) discussion, and the LC3 expression explanation with appropriate references in the paper. They should also discuss the pH results and how they might impact the study's outcomes.
Author’s response:
Thank you for your advice. We have added the results of mannitol as supplemental data and addressed the discussion regarding trehalose toxicity in the manuscript. Regarding the explanation of LC3-II expression, we included the explanation in section 2.7 "Analysis of autophagy flux". Besides, we could not compare the expression of LC3-II in different panels (control group in Figure 7A and Tre group in Figure 7B) in Figure 7, because they are evaluated separately. Additionally, the part of pH results has been revised in section 2.4 "Validation of oxygen concentrations and pH in different solutions". We hope these adjustments will meet your expectations.
Mannitol results:
(Materials and Methods, Page 4)
To exclude the effect of osmolarity, we also used mannitol (50 mM, FUJIFILM Wako Pure Chemical Corporation) and sucrose (50 mM, FUJIFILM Wako Pure Chemical Corporation) as negative controls during preconditioning and ischemia phases. Subsequently, we tested the cell viability after reperfusion by CCK-8 (Dojindo).
(Results, Page 6)
We further examined cell viability following the application of mannitol and sucrose during preconditioning and ischemia to evaluate the influence of osmolarity. No increase in cell viability was observed after treatment with mannitol and sucrose (Figure S1).
Trehalose toxicity:
(Discussion, Page 10)
In our study, we utilized a concentration of 50 mM Tre, and the CCK-8 assay results revealed no significant decrease in cell viability following a 4-hour treatment. While our concentration determination experiment did not assess subtle cellular changes, which may warrant further investigation. Trehalose is known as a cell membrane protector, with established roles in cell cryoprotection [19]. Moreover, Trehalose has been utilized in ET-Kyto solution for long-term lung preservation at a concentration of 120 mM [20]. Therefore, the potential cardioprotective effects of Trehalose, even at a concentration of 50 mM, remain promising.
Discussion about pH:
(Materials and Methods, Page 4)
Table 2 illustrates that pH levels decreased to a greater extent under ischemic conditions compared to the normal group, which suggested that the model simulated the ischemia condition, rather than solely hypoxia, during WIT and fWIT.